# Long COVID and the Autonomic Nervous System: The Journey from Dysautonomia to Therapeutic Neuro-Modulation through the Retrospective Analysis of 152 Patients

**Joseph Colombo [1],*, Michael I. Weintraub [2],*, Ramona Munoz [1], Ashish Verma [1], Ghufran Ahmad [1], Karolina Kaczmarski [1], Luis Santos [3] and Nicholas L. DePace [1]**

1   Franklin Cardiovascular, Autonomic Dysfunction and POTS Center, Sicklerville, NJ 08081, USA; rmunoz@franklincardio.com (R.M.); ashish@ashishverma.com (A.V.); ghufran.kmc@gmail.com (G.A.); kikaczmarski@gmail.com (K.K.); dovetech@erols.com (N.L.D.)
2   Department Neurology and Medicine, New York Medical College, Valhalla, NY 10595, USA
3   New Jersey Heart, Sicklerville, NJ 08081, USA; drlou214@icloud.com
*   Correspondence: jcolombo@physiops.com (J.C.); miwneuro@gmail.com (M.I.W.)

**Abstract:** Introduction. The severity and prevalence of Post-Acute COVID-19 Sequela (PACS) or long-COVID syndrome (long COVID) should not be a surprise. Long-COVID symptoms may be explained by oxidative stress and parasympathetic and sympathetic (P&S) dysfunction. This is a retrospective, hypothesis generating, outcomes study. Methods. From two suburban practices in northeastern United States, 152 long COVID patients were exposed to the following practices: (1) first, they were P&S tested (P&S Monitor 4.0; Physio PS, Inc., Atlanta, GA, USA) prior to being infected with COVID-19 due to other causes of autonomic dysfunction; (2) received a pre-COVID-19 follow-up P&S test after autonomic therapy; (3) then, they were infected with COVID-19; (4) P&S tested within three months of surviving the COVID-19 infection with long-COVID symptoms; and, finally, (5) post-COVID-19, follow-up P&S tested, again, after autonomic therapy. All the patients completed autonomic questionnaires with each test. This cohort included 88 females (57.8%), with an average age of 47.0 years (ranging from 14 to 79 years), and an average BMI of 26.9 #/in$^2$. Results. More pre-COVID-19 patients presented with sympathetic withdrawal than parasympathetic excess. Post-COVID-19, these patients presented with this ratio reversed and, on average, 49.9% more autonomic symptoms than they did pre-COVID-19. Discussion. Both parasympathetic excess and sympathetic withdrawal are separate and treatable autonomic dysfunctions and autonomic treatment significantly reduces the prevalence of autonomic symptoms. Conclusion. SARS-CoV-2, via its oxidative stress, can lead to P&S dysfunction, which, in turn, affects the control and coordination of all systems throughout the whole body and may explain all of the symptoms of long-COVID syndrome. Autonomic therapy leads to positive outcomes and patient quality of life may be restored.

**Keywords:** long COVID; parasympathetic; sympathetic; autonomic dysfunction; autonomic therapy; outcomes

## 1. Introduction

The severity and prevalence of Post-Acute COVID-19 Sequela (PACS) or long-COVID syndrome (long COVID) should not be a surprise. SARS-CoV-2 targets diverse organs and tissues after entry into the human body [1]. Long-COVID syndrome is defined as persistent symptoms beyond 12 weeks after acute COVID-19 infection [2,3]. Viruses, by inducing an inflammatory state, can damage tissue. At a cellular level, the mitochondria are susceptible to the effects of inflammation and oxidative stress [4]. Given that nerve cells, including brain cells, and heart muscle cells contain significantly more mitochondria than other cells in the body, it is to be expected that they will be the most affected by oxidative stress. The results of mitochondrial dysfunction includes primarily autonomic dysfunction (including both

parasympathetic and sympathetic (P&S)) and cardiovascular dysfunction [5]. Arguably, the first symptom of P&S dysfunction is orthostatic dysfunction [5,6]. Orthostatic dysfunction is a significant contributor to poor cardiac and cerebral perfusion (and, of course, all structures around and above the heart). Autonomic dysfunction is also induced as a result of the severity of the infection [7].

Furthermore, COVID-19 injures the lungs, reducing their ability to exchange oxygen, exacerbating the poor perfusion and resulting dysfunctions [8]. The initial respiratory compromise, due to the COVID-19 virus, on the medullary respiratory control centers (including the pre-Bötzinger complex) [9–11] may be so dramatic that P&S symptoms and signs are often overlooked or misunderstood. Respiratory pacing from the pre-Bötzinger complex involves (1) vagus nerve afferents, among other brainstem structures; (2) feedback from the COVID-19-damaged lung; (3) aortic and carotid chemo-, baro-, and vagal receptors; and (4) medullary chemoreceptors. All involving P&S nerves [9,12]. Brainstem cardiorespiratory centers (e.g., the Nucleus Tractus Solitarius, Dorsal Vagal Motor Nucleus, and Nucleus Ambiguus, all of which are autonomic nuclei) are also implicated in COVID-19 infection [13]. Furthermore, sympathetic involvement in cytokine storms [14–17] and the angiotensin system [18,19], and parasympathetic involvement in immune function [20–22], provides further evidence of P&S compromise in COVID-19 infections. Any resulting damage to these nerves further implicates P&S dysfunction in long-COVID syndrome.

Long-COVID symptoms [23] may be explained by a pro-inflammatory state with oxidative stress and P&S dysfunction [24]. This study presents the data obtained from autonomic dysfunction patients who were P&S tested and treated prior to COVID-19 infection due to other causes of autonomic dysfunction. Then, they were P&S tested and treated after surviving COVID-19 infection.

Long-COVID symptoms may be explained by a pro-inflammatory state with oxidative stress and P&S dysfunction. This is hypothesis generating.

Long COVID is characterized by parasympathetic excess and alpha-sympathetic withdrawal.

Anti-cholinergic therapy may relieve post-COVID-19 symptoms associated with parasympathetic excess. This is hypothesis generating and further trials are needed.

## 2. Methods

The data presented are from 2 suburban practices in northeastern United States (Sicklerville, NJ, USA and Valhalla, NY, USA), a cardiovascular and autonomic dysfunction clinic and a neurology clinic (respectively). From these two practices, 152 long-COVID patients from around the world who (1) had been under medical therapy for autonomic dysfunction, had been evaluated and underwent P&S testing prior to being infected with COVID-19; (2) a follow-up P&S test was conducted after autonomic therapy; (3) patients were infected with COVID-19; (4) patients were P&S tested within three months of surviving COVID-19 infection with long-COVID symptoms, typically more than the pre-COVID condition, with continued autonomic therapy adjusted to individual patients' needs; and, finally, (5) patients were follow-up P&S tested, again, after autonomic therapy. This cohort includes 88 females (57.8%). The average patient age is 47.0 years (ranging from 14 to 79 years), with an average BMI of 26.9 #/in$^2$. All the patients were tested with P&S monitoring (P&S Monitor 4.0; Physio PS, Inc., Atlanta, GA, USA) and completed a 28-symptom questionnaire (Table 1). This is a retrospective, observational, hypothesis-generating, outcomes study. All the patients permitted their data to be included in this large population study and patient data were maintained according to the HIPPA guidelines.

P&S monitoring collects EKG, respiratory activity, and BP during four challenges: (1) rest (baseline), (2) deep breathing (0.1 Hz, a parasympathetic challenge), (3) short Valsalva maneuvers (<15 s, as a sympathetic challenge), and (4) head-up postural change (stand, which is equivalent to tilt [25]). Stand is both an orthostatic challenge and a measure of the coordination between the P&S branches. With spectral analyses, these data are analyzed and independent, and simultaneous P&S activity is measured throughout the

clinical study [5]. Normal and abnormal P&S response plots are depicted in Figures 1–5, including, in order, (1) a resting baseline response (Figure 1) depicting normal and abnormal ranges; (2) a normal stand or upright posture response (Figure 2); (3) an abnormal stand response depicting alpha-sympathetic withdrawal (upon standing) which indicates orthostatic dysfunction (Figure 3); (4) an abnormal stand response depicting parasympathetic excess (upon standing) which indicates vagal excess (Figure 4); and (5) an abnormal stand response depicting parasympathetic excess with a hyperadrenergic response (upon standing) which indicates vasovagal syncope (Figure 5) [5].

**Table 1.** 28-symptom autonomic dysfunction questionnaire.

| 1. Lightheaded | 2. Fatigue | 3.Chest Pain, Palpitations | 4. Short of Breath | 5. Fainting and Near Fainting | 6. Difficulty Standing | 7. Sweat Too Much, Too Little |
|---|---|---|---|---|---|---|
| 8. Brain fog or mental cloudiness | 9. Difficulty finding words | 10. Short-term memory loss | 11. Insomnia, sleep difficulty | 12. Depression, anxiety | 13. Tension headaches | 14. Migraine, headache |
| 15. Chronic pain | 16. Coat hanger pain in neck and shoulders | 17. Pins and needs in arms/legs | 18. Numbness in hands and feet | 19. Hypermobility of joints, joints pop out | 20. Nausea, vomiting | 21. Diarrhea, constipation |
| 22. Sensory: hypersensitive to light, sound, motion, touch | 23. Sensory deficits: vision, hearing, taste, smell | 24. Cold hands or feet | 25. Ringing in ears | 26. Does hot or cold weather bother you? | 27. Hands or feet turn different colors (red, white, or blue) in cold temperatures | 28. Salivate too little, dry mouth |

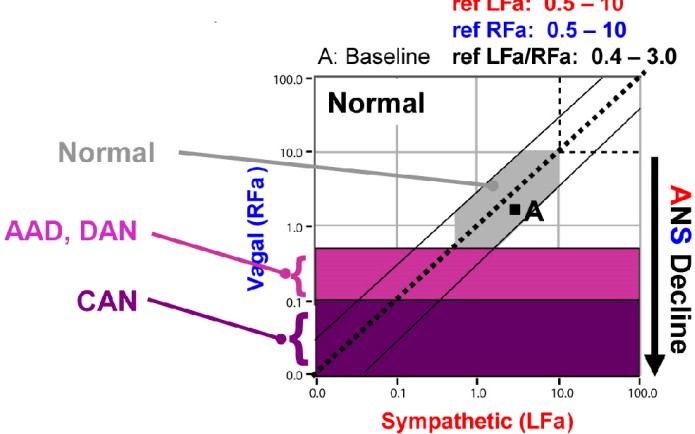

**Figure 1.** "Normal at Rest". A resting (baseline) P&S response plot depicting normal and abnormal ranges. The gray area depicts the normal response region. The purple highlighted areas depict the definitions of Advanced Autonomic Dysfunction (AAD, light purple) or Diabetic Autonomic Neuropathy (DAN, also light purple), and Cardiovascular Autonomic Neuropathy (CAN, dark purple). AAD and DAN indicate an increased morbidity risk and CAN indicates an increased mortality risk. Risk is stratified by sympathovagal balance ("LFa/RFa" = S/P). The space between the two outer diagonal lines defines a normal sympathovagal balance, regardless of the resting autonomic state. A normal sympathovagal balance normalizes the morbidity and mortality risks. Above and to the left of the upper diagonal line indicates a low sympathovagal balance, which is a resting parasympathetic excess. Below and to the right of the lower diagonal line indicate a high sympathovagal balance, which is a resting sympathetic excess.

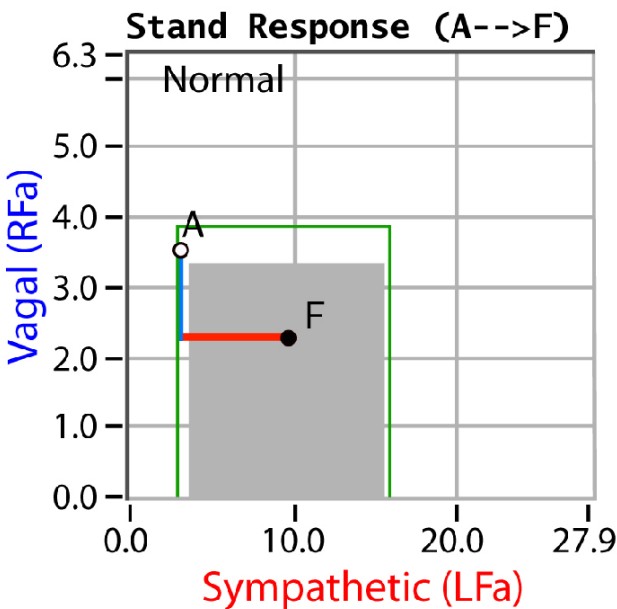

**Figure 2.** "Normal upon Standing". An example normal stand P&S response plot. Active standing is equivalent to a positive, head-up, tilt [25]. Point "A" is the patient's resting, baseline response and point "F" is the patient's stand response. In the normal stand response, the parasympathetics (the blue line) first decrease and then the (alpha-)sympathetics (the red line) increase [5].

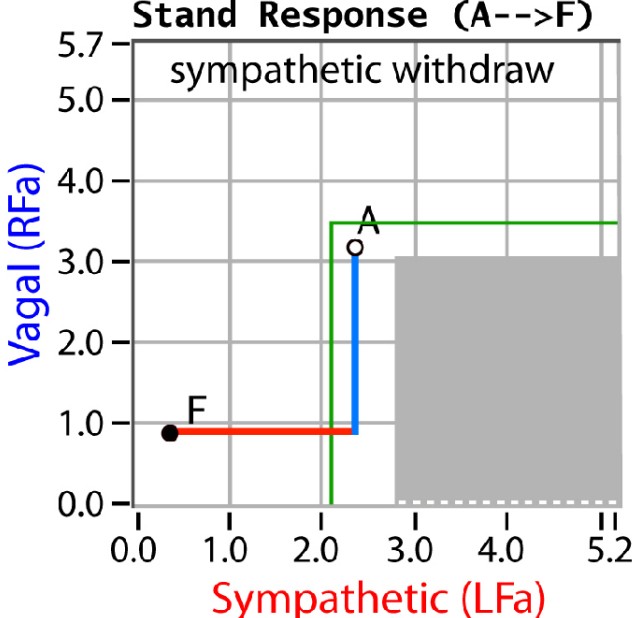

**Figure 3.** "Sympathetic Withdrawal". An example of an abnormal stand P&S response plot depicting alpha-sympathetic withdrawal. Point "A" is the patient's resting, baseline response and point "F" is the patient's stand response. Here, the parasympathetic response is normal (see Figure 2), but the sympathetic response decreases abnormally, indicating orthostatic dysfunction, possibly leading to all forms of POTS, orthostatic intolerance, orthostatic hypotension, neurogenic orthostatic hypotension, etc. [5].

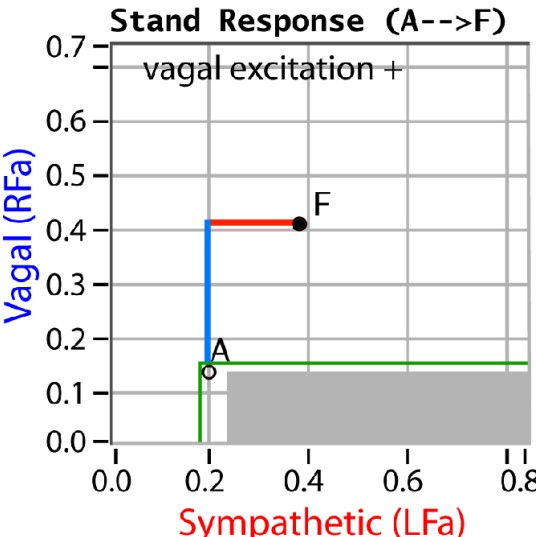

**Figure 4.** "Vagal Excitation". An example of an abnormal stand P&S response plot depicting parasympathetic excess. Point "A" is the patient's resting, baseline response and point "F" is the patient's stand response. Here, the sympathetic response is normal (see Figure 2), but the parasympathetic response increases abnormally, indicating vagal or parasympathetic excess, associated with difficult-to-control BP, blood glucose, hormone levels, or weight; difficult-to-describe pain syndromes (including CRPS); unexplained arrhythmia (palpitations) or seizures; temperature dysregulation (both the response to heat or cold and sweat responses); symptoms of depression or anxiety, ADD/ADHD, fatigue, exercise intolerance, sex dysfunction, sleep or GI disturbance, lightheadedness, cognitive dysfunction or "brain fog"; and frequent headaches or migraines. Parasympathetic excess and sympathetic withdrawal may concurrently occur, including the fact that parasympathetic excess may mask sympathetic withdrawal. This masking is indicated by an abnormal BP response to stand as compared with resting BP [5].

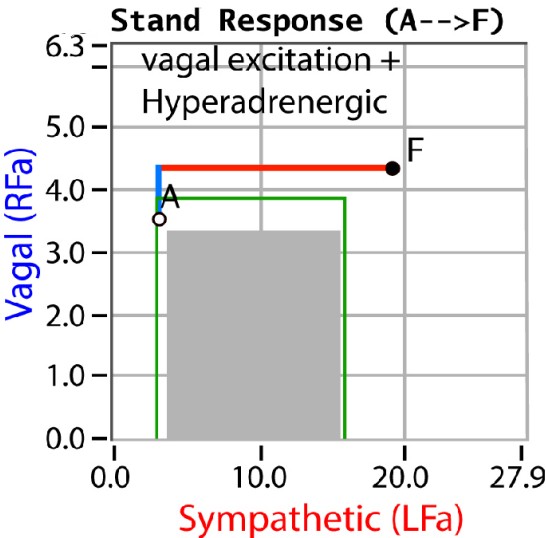

**Figure 5.** "Vagal Excitation + Hyperadrenergic". An example of an abnormal stand P&S response plot depicting parasympathetic excess with sympathetic excess. Point "A" is the patient's resting, baseline response and point "F" is the patient's stand response. Here, the parasympathetic response is abnormal (see Figure 2), as is the sympathetic response which increases too significantly, exceeding the normal area. The combination indicates vasovagal syncope. The parasympathetic excess is the vagal component, and the sympathetic excess (hyperadrenergic response) indicates the nervous system's response to syncope and the accompanying poor cerebral perfusion [5].

P&S monitoring differs from the other autonomic monitors, in that it is uniquely capable of independently and simultaneously measuring the two individual autonomic branches without assumption and approximation [26]. P&S monitoring permits follow-up testing, and includes indications for peripheral autonomic neuropathy (including Small C-Fiber Disease) [5], as well as P&S dysfunctions (including autonomic neuropathies) not detected by typical autonomic monitors, including sympathetic withdrawal (an alpha-adrenergic insufficiency upon assuming a head-up posture, associated with orthostatic dysfunction) [5,27] and parasympathetic excess (an excessive cholinergic response to a stress, as modeled by the Valsalva challenge or upon assuming a head-up posture, associated with Vagal over-reactions) [5,28].

Sympathetic withdrawal (see Figure 3) and parasympathetic excess (see Figure 4) are two of the P&S dysfunctions typically demonstrated by long-COVID patients. The others include (1) sympathetic excess with up-right posture (a beta-adrenergic response associated with syncope and pre-syncopal symptoms, see Figure 5); (2) low and (3) high sympatho-vagal balance (a measure of the ratio of sympathetic-to-parasympathetic activity at rest, see Figure 1); (4) low resting sympathetic or parasympathetic activity associated with advanced autonomic dysfunction or diabetic autonomic neuropathy if diabetic (see Figure 1); and (5) very-low resting parasympathetic activity at rest, associated with cardiovascular autonomic neuropathy (see Figure 1).

Based on their P&S test results, the patients were prescribed therapy, typically for both sympathetic withdrawal (and associated orthostatic dysfunction) and for parasympathetic excess. Therapy for sympathetic withdrawal (after ruling out vascular causes) typically included: (1) 2.5 mg, tid, of Midodrine (ProAmatine, an alpha-adrenergic antagonist); and (2) up to 600 mg, tid, of Alpha-Lipoic Acid (an antioxidant selective for nerves [29,30]). Therapy for parasympathetic excess included: (1) 10 mg, qd, of Nortriptyline (as a low-dose anti-cholinergic), and (2) up to 40 minutes of low-and-slow exercise [31]. The Pearson correlation and Student's *t*-test statistics are based on SPSS v. 20.

## 3. Results

In general, the patients reported poor health. Patients first presented (pre-COVID-19) with lightheadedness (100%), due to (1) pre-syncope (28.3%) or syncope (2.6%); (2) orthostatic dysfunction, including Postural Orthostatic Tachycardia Syndrome (POTS, 8.6%) and orthostatic intolerance or orthostatic hypotension (36.8%); or (3) excessive vagal symptoms (27.0%). Approximately, a quarter (25.7%) of the cohort first presented with anxiety-like symptoms, including palpitations and shortness of breath. Over a third (36.9%) of the cohort reported fatigue, nearly half (46.9%) reported generalized pain, including headaches and migraines, and 25.7% of the patients were diagnosed with Ehlers-Danlos Syndrome—Hypermobility (see Table 2). The prevalence of the autonomic dysfunctions are listed in Table 3. Sympathetic withdrawal is the most prevalent autonomic dysfunction pre-COVID-19, and parasympathetic excess is the most prevalent post-COVID-19.

**Table 2.** Patient demographics upon the first presentation.

| Cohort | No. | No. Female | Ave. Age | Ave. BMI | LH | Fatigue | Anxiety | Headache, Migraine | EDSh |
|---|---|---|---|---|---|---|---|---|---|
| # (%) | 152 | 88 (57.8) | 47.0 yrs | 26.9 lbs/ft$^2$ | 152 (100) | 56 (36.9) | 39 (25.7) | 71 (46.9) | 39 (25.7) |

Ave.: average; BMI: body mass index; EDSh: Ehlers-Danlos Syndrome/Hypermobility; LH: lightheadedness; and No.: number.

From the last column of Table 3, upon the initial presentation (pre-COVID-19), these autonomic dysfunction patients presented with an average of 2.34 of the 7 P&S dysfunctions listed in the first seven (7) columns of Table 3. With less than 9 months of therapy, the pre-COVID-19 patients were found with an average of 0.95 of the 7 P&S dysfunctions ($p < 0.001$). Post-COVID-19, these patients demonstrated an average of 3.67 of the 7 P&S

dysfunctions ($p = 0.004$). With less than 6 months of continued and, as needed, additional autonomic therapy, the post-COVID-19 patients were found with an average of 1.63 of the 7 P&S dysfunctions ($p = 0.003$).

**Table 3.** Percentage prevalence of the autonomic dysfunctions in long-COVID patients. See the text for details and abbreviations.

| N = 152 # (%) | SW | PE | SE | Low SB | Hi SB | AAD | CAN | Ave. Sx | Ave. ADs |
|---|---|---|---|---|---|---|---|---|---|
| Pre-COVID-19 | 69 (45.4) | 41 (27.0) | 19 (12.5) | 23 (15.1) | 41 (27.0) | 26 (17.1) | 8 (5.3) | 9.74 | 2.34 |
| Pre-COVID-19 Follow-up | 41 (27.2) | 25 (16.2) | 11 (7.5) | 14 (9.1) | 25 (16.2) | 6 (3.9) | 0 (0) | 6.25 | 0.95 |
| *p*-value | 0.023 | 0.011 | 0.009 | 0.001 | 0.002 | <0.001 | <0.001 | 0.009 | <0.001 |
| Post-COVID-19 | 55 (36.2) | 71 (46.7) | 44 (28.9) | 58 (38.2) | 69 (45.4) | 31 (20.4) | 10 (6.6) | 14.6 | 3.67 |
| Post-COVID-19 Follow-up | 33 (21.7) | 43 (28.0) | 26 (17.4) | 29 (19.1) | 31 (20.4) | 6 (3.9) | 2 (1.3) | 7.44 | 1.63 |
| *p*-value | 0.041 | 0.024 | 0.016 | 0.010 | 0.002 | <0.001 | 0.001 | 0.004 | 0.003 |

AD: average number of autonomic dysfunctions based on the seven possible dysfunctions listed as the first seven column headers of this table; AAD: Advanced Autonomic Dysfunction, an indication of morbidity risk; CAN: Cardiovascular Autonomic Neuropathy, an indication of mortality risk; PE: parasympathetic excess, an abnormal parasympathetic response to a sympathetic challenge or stress; SB: sympathovagal balance, the ratio of resting sympathetic-to-resting parasympathetic activity; SE: sympathetic excess, a beta-adrenergic response to challenge; SW: sympathetic withdrawal, an alpha-adrenergic response to positive, head-up postural change (e.g., stand); and Sx: average number of autonomic symptoms from the 28-question survey in Table 1.

From the second to last column of Table 3, at the pre-COVID-19 baseline, these patients complained of an average of 9.74 of the 28 symptoms. Upon the pre-COVID-19 follow-up, the patients' complaints were reduced, on average, to 6.25 symptoms ($p = 0.009$). Post-COVID-19, the patients complained of an average of 14.6 of the 28 symptoms ($p < 0.001$). Upon the post-COVID-19 follow-up, the patients complained of an average of 7.44 symptoms ($p = 0.004$).

COVID-19 infection returned and added to the number of (56.8% more) autonomic dysfunctions demonstrated by these 152 patients. Also, COVID-19 infection returned and increased the number of (49.9% more) associated symptoms reported by these patients. Upon follow-up testing, both pre- and post-COVID infection, all the patients reported improved outcomes, which was evidenced by the fewer P&S dysfunctions and fewer symptoms reported upon follow-up.

From Table 3, acute COVID-19 infection also reversed the order of the top two autonomic dysfunctions from sympathetic withdrawal being more predominant pre-COVID-19 to parasympathetic excess being more predominant post-COVID-19. An abnormal sympathovagal balance also become more significant. Those who also demonstrated a low sympathovagal balance (resting vagal excess) also reported more significant symptoms of depression/anxiety and fatigue. Those who demonstrated a high sympathovagal balance (resting sympathetic excess) also reported more significant symptoms of pain and hypertension. However, the high sympathovagal balance results may be biased by the number of Ehlers-Danlos Syndrome-Hypermobility patients.

## 4. Discussion

COVID-19 is documented to adversely affect the autonomic nervous system [32]. In many patients, the lingering effect on the autonomic nervous system results in what has been termed long COVID [33]. Long COVID is well documented to involve the autonomic nervous system [34–36]. Autonomic dysfunctions may be peripheral or central. In central cases, autonomic dysfunctions may be related to microglial hyperactivation inside the

brainstem autonomic centers [37]. Microglial hyperactivation is associated with PE [38]. Autonomic dysfunctions may also be highly influenced by psychological factors.

In our findings, long COVID is largely characterized by parasympathetic excess and sympathetic withdrawal. Both potentially contributing to hypoperfusion of the brain and all structures above and around the heart. Pre-COVID-19 infection, patients presented to the clinics with more sympathetic withdrawal (45.7%) than parasympathetic excess (27.0%). Post-COVID-19 infection, these patients presented with that ratio reversed (36.2% and 46.7%, respectively). The etiology of this is not well known; however, parasympathetic excess may be more prominent post-COVID-19, due to an over-active immune system, which the parasympathetics help to control and coordinate and leads to parasympathetic excess.

Given that the parasympathetic nervous system controls and coordinates the immune system, severe infections lead to excessive and prolonged parasympathetic activation in response to challenges or stressors (known as parasympathetic excess) [7], which exacerbates autonomic and cardiovascular dysfunctions. A common, and perhaps first cause of autonomic dysfunction, due to mitochondrial dysfunction and associated oxidative stress, is orthostatic dysfunction [6], resulting in poor cardiac and cerebral perfusions (and, of course, all the structures around and above the heart). Orthostatic dysfunction is caused by poor vasoconstriction due to alpha-adrenergic (sympathetic) dysfunction, known as sympathetic withdrawal [5]. Poor perfusion and dysfunction are exacerbated by the effect of COVID-19 on the lungs.

Both parasympathetic excess and sympathetic withdrawal are separate and treatable dysfunctions. As in this study, parasympathetic excess was treated, pharmaceutically, with anti-cholinergics (e.g., Nortriptyline, see the Methods Section) [31] and sympathetic withdrawal was treated, pharmaceutically, with oral vasoactives (e.g., Midodrine, see the Methods Section) [30].

Our findings demonstrate an initial worsening of autonomic dysfunction and symptoms associated with COVID-19 infection, and then, with autonomic treatment, these dysfunctions and symptoms may again be relieved. Traditionally, upon COVID-19 infection, there is a marked increase in the resting sympathetic activity and a decrease in anti-inflammatory resting parasympathetic activity [16], causing a high (resting) sympathovagal balance in all patients. However, in post-COVID-19 syndrome patients, after 12 weeks or more, our data shows that there is a significant percentage of patients that develop a parasympathetic dominance as indicated by the low (resting) sympathovagal balance. This is also indicative of increasing and prolonged parasympathetic activity. Parasympathetic activation is meant to be protective; including, since the parasympathetics are anti-inflammatory. However, prolonged and increased parasympathetic activity, especially in response to stressors, seems to exaggerate sympathetic inflammatory activity. Within this cohort, and anecdotally with the vast majority of our patients, anti-cholinergic therapy relieves parasympathetic excess. Further studies are required to elaborate whether anti-cholinergic therapy may relieve post-COVID-19 symptoms.

All symptoms of long COVID may be explained by oxidative stress and P&S dysfunction. For example, P&S dysfunction leading to orthostatic dysfunction underlies poor cerebral (including all structures above the heart) perfusion, which causes fatigue, brain-fog, cognitive and memory difficulties, sleep difficulties, and other depression-like symptoms, including "coat-hanger" pain, headaches and migraines; cranial nerve dysfunctions, including visual and auditory effects (including tinnitus), taste and smell deficits, and facial sensations due to trigeminal nerve dysfunction. P&S dysfunction may also increase BP (and may eventually lead to hypertension) as a compensatory mechanism to promote cerebral perfusion. Further decreases in cerebral perfusion may lead to "adrenaline storms", which cycle anxiety-like symptoms, including shortness of breath and palpitations which may cause chest pressure or chest pain. The effects of sympathetic withdrawal and orthostatic dysfunction are exacerbated by parasympathetic excess, which may limit or decrease the

heart rate and blood pressure, reducing cerebral perfusion. The decrease in BP is also associated with excessive vasodilation from parasympathetic excess.

If the parasympathetics increase in response to a stress (known as parasympathetic excess), the result is a secondary sympathetic excess [5]. Our findings of prolonged parasympathetic excess in long-COVID patients appears to prolong sympathetic excess responses causing more and chronic symptoms, suggesting that this may be a mechanism contributing to long-COVID syndrome.

Pharmaceutical therapy for P&S dysfunction (anti-cholinergics for parasympathetic excess [28] and oral vasoactives for sympathetic withdrawal [39]) needs to be very low to prevent additional symptoms, thereby exacerbating P&S dysfunction. From Table 3, COVID-19 significantly increases autonomic dysfunctions and the associated symptoms, and autonomic therapy significantly reduces autonomic dysfunctions and the associated symptoms. Further studies are needed, including blinded, controlled studies.

## 5. Conclusions

The current body of evidence suggests that SARS-CoV-2 can affect the nervous system in previously unexpected ways [40]. Mitochondrial damage causing oxidative stress leads to P&S dysfunction. In turn, oxidative stress and P&S dysfunction affects the control and coordination of systems throughout the body and may explain the clinical symptoms recognized as long-COVID syndrome. Autonomic therapy has been shown to provide positive outcomes and improvement in patients quality of life.

The association of anxiety, Postural Orthostatic Tachycardia Syndrome (POTS), and Chronic Fatigue Syndrome/Myalgic Encephalomyelitis (CFS/ME) with long COVID is interesting, because all are also characterized by P&S dysfunction. However, to diagnose these conditions, including long COVID, independent and simultaneous (direct) measures of P&S activity are required; the assumptions and approximations required by other autonomic tests are not typically appropriate for these patients. These direct measures are needed because both autonomic branches are involved in long COVID in different ways and must be treated separately, and may be treated simultaneously. Therapy is low and slow, and patient expectations must be properly established for optimum compliance. Follow-up testing is needed to help with compliance and ensure that therapy is properly titrated to the individual patient. Based on all of this, positive outcomes are realized and patient quality of life may be restored. While this study serially followed patients with underlying autonomic dysfunctions pre- and post-COVID-19, future studies should assess the effects of autonomic functions on normal subjects pre- and post-COVID-19.

**Author Contributions:** Conceptualization and writing, J.C. and N.L.D.; formal analysis, N.L.D.; investigation, J.C.; data collection, R.M. and L.S.; data mining, A.V., G.A. and K.K.; supervision, M.I.W. All authors have read and agreed to the published version of the manuscript.

**Funding:** This research received no external funding.

**Institutional Review Board Statement:** All patients permitted their data to be included in this large population study and patient data were maintained according to HIPPA guidelines.

**Informed Consent Statement:** Patient consent was waived due to all patients attending these clinics are informed that their data may be used for large population clinical studies, unless the patient objects. None of the 152 patients included in this study objected.

**Data Availability Statement:** All data are HIPAA protected and available upon request.

**Conflicts of Interest:** Only Colombo has a conflict of interest as Co-Founder and Sr. Medical Director of Physio PS, Inc. No other individual associated with this manuscript has a conflict of interest.

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
