# Peer review of "Long COVID and the Autonomic Nervous System: The Journey from Dysautonomia to Therapeutic Neuro-Modulation through the Retrospective Analysis of 152 Patients"

_neurosci, doi:10.3390/neurosci3020021_

Round 1
Reviewer 1 Report
The manuscript presented from Colombo et al., is interesting. However there are many critical points to improve:
- The authors should improve the introduction
- The authors in the text mentioned see figure 1,2,3... but these figures are not presented in the manuscript. Where are they?
- The authors should explain in detail the methods used.
- The authors should include a Table to summarize the results
- The style of the reference should be ACS.
Author Response
Reviewer #1: The manuscript presented from Colombo et al., is interesting. However there are many critical points to improve:
- The authors should improve the introduction, providing sufficient background and include all relevant references
- We have included additional relevant references, however, taking care to be limited as, another reviewer commented on the excessive number of references. In doing so we believe we have improved the introduction as recommended.
- The authors in the text mentioned see figure 1,2,3... but these figures are not presented in the manuscript. Where are they?
- The figures and figure legends and Tables were submitted in the original manuscript, the figures were at the end of the manuscript, just before the references. Upon resubmission we will ensure figures and tables are included.
- The authors should explain in detail the methods used.
- We have modified the Methods to provide all methodological details in this section.
- The authors should include a Table to summarize the results.
- We have modified Table 3 to also include a summary of the results.
- The style of the reference should be ACS.
- Thank you.
Reviewer 2 Report
Cov2 is a RNA virus that does not interact with DNA nether nuclear, nor mitochondrial. It is cytopathic virus that produces acute respiratory disease.
There is no neuropathological evidence the the virus significantely reproduces in the CNS, nor in the paripheral nervous system.
The whole article is based on conjectures.
Autonomic alterations may be related to microglial hyperactivation inside the brainstem autonomic centers, and probably they are not due to a direct viral invasion. Also, they may be highly influenced by psychological factors that are never mentioned in the paper.
Author Response
Cov2 is a RNA virus that does not interact with DNA nether nuclear, nor mitochondrial. It is cytopathic virus that produces acute respiratory disease.
- Thank you we have modified the text to reflect this comment.
- There is no neuropathological evidence the the virus significantely reproduces in the CNS, nor in the paripheral nervous system.
- Thank you we have modified the text to reflect this comment.
- The whole article is based on conjectures.
- We have removed the generalist terms and made appropriate notes as to the hypothesis generating nature of the manuscript.
- Autonomic alterations may be related to microglial hyperactivation inside the brainstem autonomic centers, and probably they are not due to a direct viral invasion. Also, they may be highly influenced by psychological factors that are never mentioned in the paper.
- Thank you for this point, it has been referenced
Reviewer 3 Report
Data should be better shown, I cannot see the figures and the tables the authors refer to.
The statistical test used must be explained. It is not clear to which comparison are referring the p values.
There are too many references, try to reduce the redundant ones.
The discussion is too long, since, after the hypothesis on covid19 autonomic dysfunction mechanism, it is too much focused on general autonomic dysfunctions.
Author Response
Data should be better shown, I cannot see the figures and the tables the authors refer to.
- We have modified Table 3 to better show the data. The figures and figure legends and Tables were submitted in the original manuscript, the figures were at the end of the manuscript, just before the references. Upon resubmission we will ensure figures and tables are included.
- The statistical test used must be explained. It is not clear to which comparison are referring the p values.
- We specified that the Student T-test is the comparison upon which the p-values are based.
- There are too many references, try to reduce the redundant ones.
- Thank you we have removed several redundant references, however another commented that more references were needed. We attempted .to strike a balance.
- The discussion is too long, since, after the hypothesis on covid19 autonomic dysfunction mechanism, it is too much focused on general autonomic dysfunctions.
- We have shortened the discussion primarily by summarizing the last two paragraphs.
Round 2
Reviewer 1 Report
The authors improved the quality of the manuscript, I would propose the acceptance.
Author Response
Thank you for your kind comment.
Reviewer 2 Report
The answers to the issues we raised are satisfactory and the whole paper in much more focused and clear. To reinforce the importance of microglial activation in the brainstem we suggest to quote the following paper at the beginning of the discussion:
Poloni TE, Medici V, Moretti M, Visonà SD, Cirrincione A, Carlos AF, Davin A, Gagliardi S, Pansarasa O, Cereda C, Tronconi L, Guaita A, Ceroni M (2021) COVID-19-related neuropathology and microglial activation in elderly with and without dementia. Brain Pathol 31:e12997; https://doi.org/10.1111/bpa.12997
Author Response
Thank you for your kind comments. We have added the reference as requested.